# GENFACETALK: GENERALIZABLE ONE-SHOT TALKING-HEAD GENERATION FOR DIVERSE STYLES

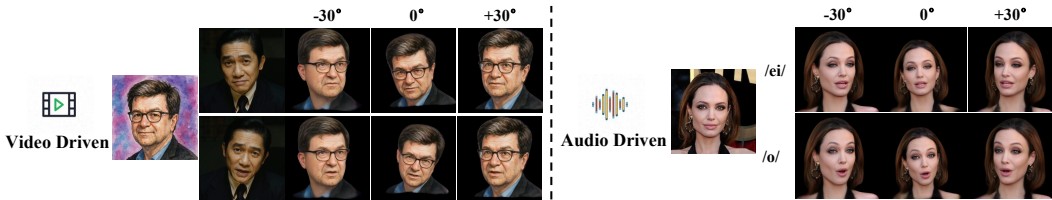

Figure 1: Example results of our *GenFaceTalk* for 3D talking-head generation, supporting both audio- and video-driven scenarios without subject-specific training.

## ABSTRACT

One-shot 3D talking-head synthesis aims to generate realistic 3D facial animations from a single portrait image, driven by audio or video inputs. While recent advances in 3D-aware generation, particularly 3D Gaussian Splatting, have enabled high-fidelity modeling and real-time rendering, existing methods still struggle with critical challenges: (i) accurate identity preservation without multi-view supervision, and (ii) producing temporally coherent animations free from jitter. We propose *GenFaceTalk*, a novel end-to-end one-shot 3DGS-based framework supporting both audio- and video-driven scenarios without subject-specific training. The core insight of GenFaceTalk is to directly predict motion-disentangled FLAME parameters from the driving video, distilling the reliance on pre-trained 3D face reconstruction and sliding-window-based smoothing into the encoder during training. This design removes the need for face reconstruction at inference, yielding temporally consistent animation while preserving identity and fine-grained facial details. We further introduce a joint learning strategy that integrates FLAME-based motion priors with hierarchical appearance features from the source, guiding 3DGS learning in a spatially aligned and identity-aware manner. Our framework generalizes across diverse facial styles, including artistic and animal faces. Experiments demonstrate that GenFaceTalk outperforms state-of-the-art baselines in visual fidelity, temporal stability, identity preservation, and cross-domain generalization.

## 1 INTRODUCTION

One-shot talking-head generation, which reconstructs and animates a human head from a single portrait using driving signals such as audio or video, has gained increasing attention due to its broad applications in virtual reality (Chen et al., 2025a), online conferencing (Wang et al., 2021), and interactive agents (Zhou et al., 2025). The primary challenge lies in faithfully preserving the visual identity of the source while accurately capturing motion and facial expressions consistent with the driving input. Existing methods can be broadly categorized into 2D-based and 3D-based approaches. While 3D-based methods typically produce more realistic results and support a wider range of applications, they often suffer from appearance inconsistency across views, especially under large pose variations. This challenge becomes more pronounced in one-shot settings, where the avatar must be reconstructed from a single image.

Recent studies have applied Neural Radiation Fields (NeRFs) to 3D talking head synthesis (Li et al., 2023; Ye et al., 2024). However, NeRF-based methods generally require subject-specific training and are computationally expensive, limiting their practicality for real-time applications. More recently, 3D Gaussian Splatting (3DGS) has emerged as a promising alternative, with some approaches achieving real-time performance (Chu & Harada, 2024; Gong et al., 2025). Despite these advances, current 3DGS-based methods still rely on identity-specific training and fail to generalize to unseen subjects (Cho et al., 2024). Moreover, they are not one-shot, usually needing multi-view images or minutes of monocular video for training (Li et al., 2024).

Few works have addressed one-shot talking-head synthesis with 3DGS. GAGAvatar (Chu & Harada, 2024) models identity-specific geometry and facial details but is limited to video-driven scenarios without audio support. The most relevant work, MGGTalk (Gong et al., 2025), generalizes to unseen identities from monocular video without fine-tuning, but its reliance on semantic parsing to separate the head from the torso/background often introduces boundary artifacts. In contrast, our method models the head as a unified entity, yielding more coherent and visually consistent results.

In this paper, we propose *GenFaceTalk*, a novel one-shot talking-head generation framework based on 3D Gaussian Splatting. Given a single portrait image, GenFaceTalk reconstructs a personalized 3D head avatar and animates it using audio or video input, without subject-specific training (see Fig. 1). One-shot talking-head generation with 3DGS faces several key challenges: **i)** Identity-preserving head modeling. Constructing a high-quality 3D head avatar from a single image is inherently ill-posed due to the absence of multi-view supervision. This often leads to poor convergence of 3D Gaussians, resulting in degraded reconstruction quality and loss of appearance details, especially under large head motions. **ii)** Temporal coherence in animation. Without effective temporal modeling, 3DGS-based synthesis suffers from frame-to-frame jitter and instability. Ensuring smooth, temporally consistent animation remains challenging in the one-shot setting. **iii)** Generalization across identities and domains. Existing methods often rely on subject-specific tuning or predefined facial models, limiting their ability to generalize. In contrast, GenFaceTalk learns a unified representation that generalizes to unseen identities, stylized portraits, and even non-human faces.

To address these challenges, GenFaceTalk's core idea is to directly map video frames to FLAME parameters using a lightweight encoder, distilling the reliance on pre-trained 3D face reconstruction models (e.g., DECA (Feng et al., 2021)) and sliding-window-based smoothing into the encoder during training. This design enables more accurate, adaptable, and temporally coherent avatar animation. Specifically, we introduce a dedicated image encoder that directly predicts FLAME parameters from the driving video. These parameters represent a motion-disentangled facial structure and are combined with appearance features extracted from the source portrait to construct a coarse FLAME mesh sequence. The mesh vertices provide strong priors on head pose and facial expression, guiding the learning of 3D Gaussian positions. In parallel, both image-level and hierarchical features from the source encode global and fine-grained appearance cues, which are fused with driving pose features to supervise 3DGS learning in a motion- and identity-aware manner. Our approach produces more accurate, generalizable, and temporally stable FLAME representations, resulting in substantial improvements in both reconstruction fidelity and animation quality.

These designs offer three key advantages over existing 3DGS-based one-shot methods: **i) Reduced temporal jitter.** Unlike prior approaches that extract FLAME parameters independently for each frame using 3DMM-based models, GenFaceTalk directly regresses temporally consistent FLAME codes from driving video, producing smoother motion and more stable animation. **ii) Fully end-to-end framework.** In contrast to methods like GAGAvatar (Chu & Harada, 2024), which rely on offline steps such as per-frame 3DMM fitting or camera transformation estimation, our approach is trained and deployed end-to-end, enabling more flexible and generalizable synthesis. **iii) Unified modeling of diverse facial styles.** While most existing methods (Gong et al., 2025) separate the face, neck, and background and are limited to conventional human faces, our method learns a holistic portrait representation from video, allowing identity-preserving modeling and animation of non-human avatars (e.g., cats, dogs) and stylized portraits.

Our main contributions can be summarized as follows:

- An end-to-end one-shot 3DGS-based framework for talking-head synthesis. Our approach directly predicts FLAME parameters, enabling temporally coherent animation without relying on 3D face reconstruction models or offline post-processing at inference time.

- Motion- and identity-aware 3DGS learning. We introduce a joint learning strategy that fuses motion priors from FLAME with hierarchical appearance features from the source to guide the 3D Gaussian representation in a spatially aligned and identity-preserving manner.
- Strong generalization across identities and visual domains. GenFaceTalk extends beyond conventional human faces to handle diverse facial styles, achieving superior visual fidelity, temporal stability, and domain generalization over baseline models.

## 2 RELATED WORK

### 2.1 ONE-SHOT TALKING-HEAD GENERATION

Early one-shot talking-head methods mainly adopt 2D-based techniques (Prajwal et al., 2020; Ma et al., 2023), focusing on audio-lip synchronization via GAN (Zhang et al., 2023; Sun et al., 2024) or diffusion models (Chen et al., 2025c; Wang et al., 2025a; Lin et al., 2025). These methods typically disentangle and align motion features between driving signals and source identities. However, lacking 3D supervision often leads to artifacts such as appearance inconsistencies, blurring, and unrealistic head poses, particularly under large head rotations or significant motion. While diffusion-based approaches offer higher visual fidelity, their computational cost limits practical deployment.

In contrast, 3D-based talking-head synthesis usually requires person-specific training on multi-view images or several minutes of monocular video, restricting scalability. One-shot 3D generation remains challenging due to the difficulty of ensuring 3D consistency in reconstruction and animation from a single image. Prior works have explored 3D Morphable Models (3DMM)-based (Huang et al., 2022; Xie et al.) and Neural Radiance Fields (NeRF)-based (Guo et al., 2021; Li et al., 2023) solutions, but they often suffer from inaccurate geometry, slow convergence, and high computational cost. In this work, we leverage the recent 3DGS framework to enable one-shot talking-head generation, achieving both high visual realism and efficient rendering.

### 2.2 TALKING-HEAD GENERATION WITH 3DGS

3DGS has recently gained attention for its high rendering efficiency and visual quality. Several works have explored 3DGS for one-shot face reenactment (Chu & Harada, 2024; Wu et al., 2024), focusing on head construction and animation from a single image. For example, GAGAvatar (Chu & Harada, 2024) incorporates 3DMM priors into the 3DGS framework via a dual-lifting strategy, while LAM (He et al., 2025) builds a canonical Gaussian avatar by querying FLAME keypoints over multi-scale image features. While these methods target video-driven animation, our GenFaceTalk extends this line to the more challenging audio-driven one-shot scenario.

Several recent methods apply 3DGS to audio-driven talking-head generation, but most require subject-specific training and are not truly one-shot (Cho et al., 2024; Li et al., 2024). PointTalk (Xie et al., 2025) initializes a coarse Gaussian head from a random point cloud and progressively refines it through a static Gaussian field. InsTaG (Li et al., 2025) learns an identity-agnostic motion field from large-scale video pretraining and adapts it to novel subjects via motion-aligned adaptation, enabling synthesis from short monocular videos. Most related to our work, MGGTalk (Gong et al., 2025) generalizes to unseen identities without personalized re-training, but relies on post-processing such as 3DMM estimation. In contrast, GenFaceTalk adopts a fully end-to-end pipeline and generalizes across diverse facial styles, supporting both human and non-human avatars.

### 2.3 CROSS-DOMAIN TALKING-HEAD GENERATION

Recent works have moved beyond identity preservation to explore cross-domain generalization in talking-head generation (Wang et al., 2025b). The goal is to synthesize facial animations for diverse identities and visual styles, ranging from natural human faces to animals (e.g., cats and dogs) and artistic or stylized portraits. Existing approaches typically adopt domain-specific motion estimators and rely on explicit motion alignment (Gong et al., 2023). For instance, AnyTalk (Wang et al., 2025b) introduces a canonical motion space to facilitate motion transfer across domains. However, these methods are mostly 2D-based and depend on hand-crafted alignment strategies. In contrast, our framework leverages a unified head construction strategy to generalize across heterogeneous facial domains without requiring explicit alignment.

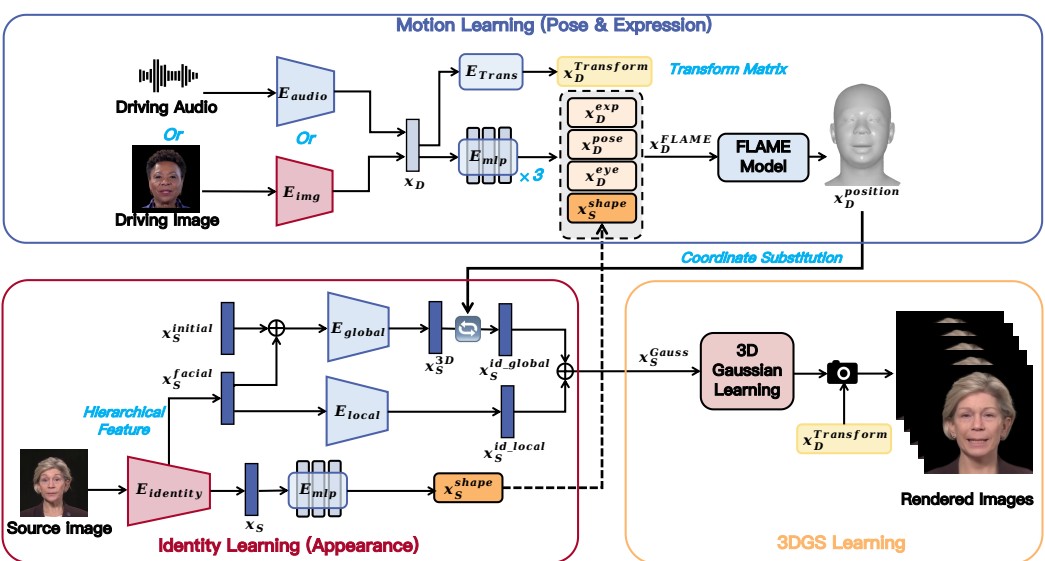

Figure 2: Overview of GenFaceTalk, consisting of a motion learning module, an identity learning module, and a 3DGS rendering module. Given a driving frame (or audio), the model predicts FLAME parameters to capture head motion, which are fused with source shape and appearance features to animate the portrait with temporally coherent pose, expression, and high visual fidelity.

## 3 METHOD

One-shot talking-head generation with 3DGS suffers from unstable convergence, temporal jitter, and poor generalization beyond human faces, since most methods rely on per-frame 3DMM reconstructions. To overcome these issues, we introduce GenFaceTalk, an end-to-end framework with three components (Fig. 2): **1) a motion learning module** (Sec. 3.1): an image encoder that directly predicts motion-disentangled FLAME parameters from driving images, enabling fine-grained control over head pose and facial expressions in a temporally consistent manner. **2) an identity learning module** (Sec. 3.2): a dual-branch representation learner that extracts hierarchical appearance features and enforces motion-aware identity consistency, ensuring efficient source identity construction. **3) a 3DGS-based renderer** (Sec. 3.3): real-time rendering that synthesizes talking-head videos conditioned on motion and identity cues.

### 3.1 MOTION LEARNING

Existing methods typically rely on frame-by-frame FLAME fitting through external 3DMM-based reconstruction pipelines, which often lead to unstable convergence and temporal jitter. Moreover, these pipelines are tailored for regular human faces and struggle to generalize across non-human or stylized domains, such as cartoons and animal avatars. We address this by introducing a lightweight encoder that directly infers motion-disentangled FLAME parameters from driving videos, removing reliance on offline 3D face reconstruction models and sliding-window smoothing during training and inference while performing face reconstruction only once during preprocessing, thereby enabling temporally coherent animation across diverse domains.

Given a driving image $D$, we first extract a global visual feature $x_D$ using an encoder $E_{img}(\cdot)$:

$$x_D = E_{img}(D) \tag{1}$$

For audio-driven scenarios, we replace the image encoder with an audio encoder $E_{audio}$, and define $x_D = E_{audio}(D)$ accordingly.

To obtain a motion-disentangled representation, we introduce three specialized MLP-based sub-encoders $E_{mlp}$: an expression extractor $E_{exp}$, a pose extractor $E_{pose}$, and an eye extractor $E_{eye}$. Each sub-encoder projects the global feature $x_D$ into a dedicated subspace:

$$x_D^m = E_m(x_D), \quad m \in \{exp, pose, eye\} \tag{2}$$

Meanwhile, a shape extractor $E_{shape}$ is applied to the source image $x_S$ to extract identity-specific shape features:

$$x_S^{shape} = E_{shape}(x_S) \tag{3}$$

We then concatenate the motion and shape features to obtain the complete FLAME code vector:

$$x_D^{FLAME} = [x_D^{exp}, x_D^{pose}, x_D^{eye}, x_S^{shape}], \tag{4}$$

$$f = M(x_D^{FLAME}) \tag{5}$$

where $M(\cdot)$ denotes the FLAME-based head construction process. The resulting head mesh $f$ comprises 5,023 vertices, whose coordinates $x_D^{position}$ encode the target head pose and facial geometry, which are subsequently used to guide Gaussian positioning in the identity learning stage (Sec. 3.2).

To enable end-to-end camera estimation, we also introduce a lightweight transformation encoder $E_{Trans}$ to predict the camera extrinsic parameters from the driving feature:

$$x_D^{Transform} = E_{Trans}(x_D) \tag{6}$$

which eliminates the need for explicit camera pose estimation at inference time.

## 3.2 IDENTITY LEARNING

Inspired by GAGAvatar (Chu & Harada, 2024), which constructs source identity using human head topology priors provided by 3DMM models and source image features, we propose a unified identity representation that fuses hierarchically extracted source image features with FLAME parameters. Unlike GAGAvatar, which relies on a fixed pre-trained DINOv2 (Oquab et al., 2023) backbone, our approach employs a hierarchical feature extraction pipeline that captures both structural identity cues and fine-grained appearance details. This richer representation enables more robust motion transfer and better identity preservation, particularly under large pose changes and expressive facial motions.

Specifically, we employ an identity encoder $E_{identity}(\cdot)$, which shares the same architecture as the image encoder $E_{img}$ used in the motion learning module (see Eq. 1), to extract hierarchical features from the source image $S$:

$$x_S^{facial} = E_{identity}(S) \tag{7}$$

Let $x_S^{initial}$ denote the initial 3D Gaussian parameters for the source identity. We concatenate these parameters with the extracted hierarchical features and feed them into a global encoder $E_{global}$ to obtain geometry-aware identity embeddings:

$$x_S^{3D} = E_{global}(x_S^{initial} \oplus x_S^{facial}) \tag{8}$$

To animate the avatar, we incorporate the driving motion features $x_D^{position}$, extracted by the Motion Learning module (Sec. 3.1), by replacing the spatial coordinates of the source Gaussians $x_S^{position}$:

$$x_S^{id\_global} = \text{Replace}(x_S^{3D}, x_S^{position}, x_D^{position}) \tag{9}$$

where Replace($\cdot$) denotes substituting the Gaussian positions of the source with those derived from the driving image.

To enhance appearance fidelity, we follow GAGAvatar (Chu & Harada, 2024) and adopt a local feature encoder $E_{local}$ to extract fine-grained identity features:

$$x_S^{id\_local} = E_{local}(x_S^{facial}) \tag{10}$$

The global and local encoders, $E_{global}$ and $E_{local}$, are convolutional networks designed to encode complementary identity representations.

Finally, the complete 3D Gaussian representation is obtained by fusing the global and local features:

$$x_S^{Gauss} = x_S^{id\_global} \oplus x_S^{id\_local} \tag{11}$$

This hybrid representation, which combines hierarchical appearance cues from the source and motion priors from the driving input, enables high-fidelity and temporally stable animation while preserving identity across diverse expressions and poses.

### 3.3 3DGS Learning

Our rendering pipeline is built upon the Gaussian rasterization framework proposed in (Kerbl et al., 2023). Specifically, the set of 3D Gaussians $x_S^{Gauss}$, produced by the identity learning module, is rendered via volumetric splatting, enabling the synthesis of high-quality and visually coherent talking-head frames.

Unlike prior 3DGS-based approaches that rely on offline preprocessing to estimate and smooth per-frame camera transformations, GenFaceTalk integrates camera pose estimation into the training process. A lightweight encoder predicts smoothed transformation matrices $x_D^{Transform}$, enabling the model to generalize during inference without explicit pose estimation.

The final rendered frame is computed as:

$$\mathbf{I} = \text{Splat}(x_S^{Gauss} \mid x_D^{Transform}) \tag{12}$$

where $\mathbf{I}$ denotes the synthesized talking-head image.

### 3.4 Loss Functions

During training, we randomly sample two frames from the same video, using one as the source and the other as both the driving and target frame. To ensure alignment between the output and target, we employ two loss functions: pixel-level L1 loss $\mathcal{L}_1$ and perceptual loss $\mathcal{L}_p$ (Johnson et al., 2016).

**FLAME code supervision.** To guide the learning of facial motion representations, we introduce a soft FLAME code loss $\mathcal{L}_{\text{FLAME}}$. The ground-truth FLAME parameters $x_{GT}^{FLAME}$ are obtained from a pre-trained 3DMM estimator in an offline manner. The predicted parameters $x_D^{FLAME}$ are then supervised using an $\ell_2$ loss:

$$\mathcal{L}_{FLAME} = \sum \left| x_D^{FLAME} - x_{GT}^{FLAME} \right|_2^2 \tag{13}$$

**Camera transform supervision.** Unlike previous methods that rely on offline preprocessing to estimate and smoothing to per-frame camera transformations, our approach introduces an end-to-end framework that eliminates the need for any post-processing during inference. To achieve this, we incorporate a transform loss $\mathcal{L}_{Transform}$ to directly supervise the learning of camera motion. During training, offline-estimated camera poses $x_{GT}^{Transform}$ are used as supervision for the predicted transforms $x_D^{Transform}$:

$$\mathcal{L}_{Transform} = |x_D^{Transform} - x_{GT}^{Transform}|_2^2 \tag{14}$$

The overall training objective is as follows:

$$\mathcal{L} = \mathcal{L}_1 + \lambda_p \mathcal{L}_p + \lambda_{FLAME} \mathcal{L}_{FLAME} + \lambda_{Transform} \mathcal{L}_{Transform} \tag{15}$$

with $\lambda_p = 0.01$ and $\lambda_{Transform} = 5$ in our experiments. To mitigate potential biases in the FLAME parameters, we set the loss weight $\lambda_{FLAME}$ to 5 for the first 20,000 training iterations and reduce it to 1 thereafter. This allows the model to leverage FLAME priors in the early stage while relying more on perceptual and L1 losses later, reducing FLAME-induced errors and improving generalization across diverse identities.

## 4 Experiment

### 4.1 Experimental Settings

**Datasets.** We evaluate GenFaceTalk on two datasets: **i) the testing set of the VFHQ dataset** (Xie et al., 2022) (i.e., VFHQ-Test), containing 50 clips of diverse interview scenarios featuring subjects from over 20 countries. Each clip is at $512 \times 512$ resolution with a frame rate of 25 FPS. **ii) the High-Definition Talking Face (HDTF) dataset** (Zhang et al., 2021), comprising 362 talking-face videos totaling approximately 15.8 hours. We follow the original preprocessing pipeline (Zhang et al., 2021), face localization via landmark detection and frame resizing to $512 \times 512$.

Table 1: Quantitative results of audio-driven methods on the VFHQ and HDTF datasets. We color the best methods.

| Method | VFHQ (Xie et al., 2022) | | | | | HDTF (Zhang et al., 2021) | | | | |
|---|---|---|---|---|---|---|---|---|---|---|
| | PSNR ↑ | SSIM ↑ | LPIPS↓ | FVD ↓ | Sync ↑ | PSNR ↑ | SSIM ↑ | LPIPS↓ | FVD ↓ | Sync ↑ |
| Wav2Lip | 20.15 | 0.840 | 0.275 | 240.91 | 7.40 | 19.54 | 0.662 | 0.283 | 219.89 | 7.52 |
| SadTalker | 17.21 | 0.787 | 0.411 | 215.57 | 8.28 | 16.54 | 0.765 | 0.390 | 199.84 | 8.08 |
| StyleHeat | 16.07 | 0.646 | 0.230 | 358.61 | 6.53 | 15.46 | 0.532 | 0.467 | 417.59 | 6.48 |
| EchoMimic | 23.61 | 0.656 | 0.201 | 212.45 | 7.37 | 21.48 | 0.690 | 0.208 | 188.07 | 7.14 |
| Hallo3 | 21.26 | 0.785 | 0.2478 | 238.08 | 7.58 | 20.62 | 0.5784 | 0.2773 | 163.54 | 8.20 |
| OmniAvatar | 19.02 | 0.659 | 0.2675 | 251.32 | 7.33 | 17.43 | 0.6113 | 0.2505 | 242.29 | 7.95 |
| Real3DPortrait | 17.97 | 0.812 | 0.353 | 281.48 | 8.44 | 16.82 | 0.770 | 0.383 | 365.85 | 8.38 |
| **Our Method** | 24.98 | 0.875 | 0.1541 | 144.58 | 8.32 | 22.52 | 0.838 | 0.194 | 159.44 | 8.63 |

Table 2: Quantitative results of video-driven methods on the VFHQ and HDTF datasets. We color the best methods.

| Method | VFHQ (Xie et al., 2022) | | | | | | | HDTF (Zhang et al., 2021) | | | | | | |
|---|---|---|---|---|---|---|---|---|---|---|---|---|---|---|
| | Self-Reenactment | | | | | Cross-Reenactment | | Self-Reenactment | | | | | Cross-Reenactment | |
| | PSNR ↑ | SSIM ↑ | FVD ↓ | AED↓ | APD↓ | AED↓ | APD↓ | PSNR ↑ | SSIM ↑ | FVD ↓ | AED↓ | APD↓ | AED↓ | APD↓ |
| StyleHeat | 17.15 | 0.685 | 423.34 | 0.1095 | 1.16E-2 | 0.1944 | 5.04E-2 | 16.55 | 0.649 | 422.27 | 0.0893 | 3.29E-2 | 0.1355 | 6.77E-2 |
| LivePortrait | 22.34 | 0.647 | 254.30 | 0.0461 | 5.39E-3 | 0.1757 | 4.29E-2 | 19.44 | 0.638 | 241.71 | 0.0355 | 1.25E-2 | 0.1821 | 3.11E-2 |
| Synergizer | 24.15 | 0.780 | 155.76 | 0.0168 | 4.36E-3 | 0.0794 | 2.73E-2 | 19.44 | 0.638 | 241.71 | 0.0186 | 1.73E-2 | 0.1497 | 4.36E-2 |
| Real3DPortrait | 22.92 | 0.771 | 257.14 | 0.0351 | 7.05E-3 | 0.1387 | 2.53E-2 | 21.67 | 0.604 | 364.82 | 0.0481 | 5.09E-3 | 0.1925 | 5.01E-2 |
| Portrait4D-v2 | 22.99 | 0.534 | 236.38 | 0.304 | 2.33E-2 | 0.0959 | 2.39E-2 | 21.62 | 0.685 | 266.83 | 0.0597 | 4.82E-2 | 0.1152 | 1.08E-2 |
| GAGAvatar | 21.17 | 0.806 | 169.80 | 0.0161 | 1.88E-3 | 0.0840 | 2.19E-2 | 20.42 | 0.746 | 212.31 | 0.0299 | 3.69E-3 | 0.0971 | 1.59E-2 |
| **Our Method** | 25.96 | 0.895 | 103.45 | 0.0106 | 1.93E-3 | 0.0714 | 2.01E-2 | 23.09 | 0.829 | 116.02 | 0.0196 | 3.13E-3 | 0.0998 | 1.49E-2 |

We conduct both audio-driven and video-driven experiments. **i) Audio-driven:** The first frame of each clip is used as the source image. Since VFHQ lacks audio, we augment it by aligning selected YouTube videos with matched audio tracks for evaluation. **ii) Video-driven:** The first frame is used as the source, and the remaining frames serve as driving and target images for same-identity reenactment. For cross-identity reenactment, we use 10 in-the-wild videos from VFHQ's training set (each 60s). No subject overlap exists between the training and test sets. We also include qualitative results on unconstrained in-the-wild examples to demonstrate generalization.

**Evaluation metrics.** Following prior works (Chu & Harada, 2024; Gong et al., 2025), we evaluate GenFaceTalk across four dimensions: **1) Image quality:** PSNR, SSIM, and LPIPS; **2) Video quality:** Fréchet Video Distance (FVD) (Unterthiner et al., 2018); **3) Lip-sync accuracy (audio-driven):** SyncNet Confidence Score (Sync) (Chung & Zisserman, 2016); **4) Motion consistency (video-driven):** Average Expression Distance (AED) and Average Pose Distance (APD).

**Implementation details.** We implemented our framework in Pytorch. All experiments were conducted with 1 NVIDIA A100 GPUs. The model is optimized using the Adam optimizer with a batch size of 10 and a learning rate of 1e-4.

## 4.2 COMPARE WITH STATE-OF-THE-ART METHODS

### 4.2.1 BASELINES

We compare GenFaceTalk with recent state-of-the-art talking-head generation methods under both audio-driven and video-driven settings. **(1) Audio-driven baselines** include 2D methods such as **Wav2Lip** (Prajwal et al., 2020), **SadTalker** (Zhang et al., 2023), **StyleHeat** (Yin et al., 2022), **EchoMimic** (Chen et al., 2025b), **Hallo3** (Cui et al., 2025), and **OmniAvatar** (Gan et al., 2025), as well as 3D rendering-based approaches like **Real3DPortrait** (Ye et al., 2024). **(2) Video-driven baselines** cover 2D methods (e.g., **StyleHeat** (Yin et al., 2022), **LivePortrait** (Guo et al., 2024) and **Synergizer** (Zhao et al., 2025) (short for Synergizing Motion and Appearance), 3D NeRF-based methods including **Real3DPortrait** (Ye et al., 2024), and **Portrait4D-v2** (Deng et al., 2024), and 3DGS-based methods such as **GAGAvatar** (Chu & Harada, 2024). All baselines are evaluated using their official implementations.

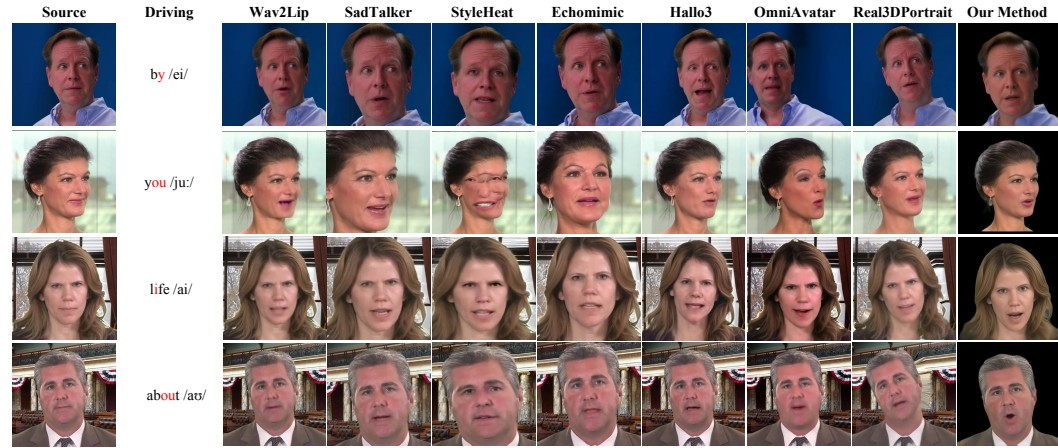

Figure 3: Audio-driven talking-head generation results. Comparisons with state-of-the-art methods on VFHQ (rows 1–2) and HDTF (rows 3–4) datasets are shown.

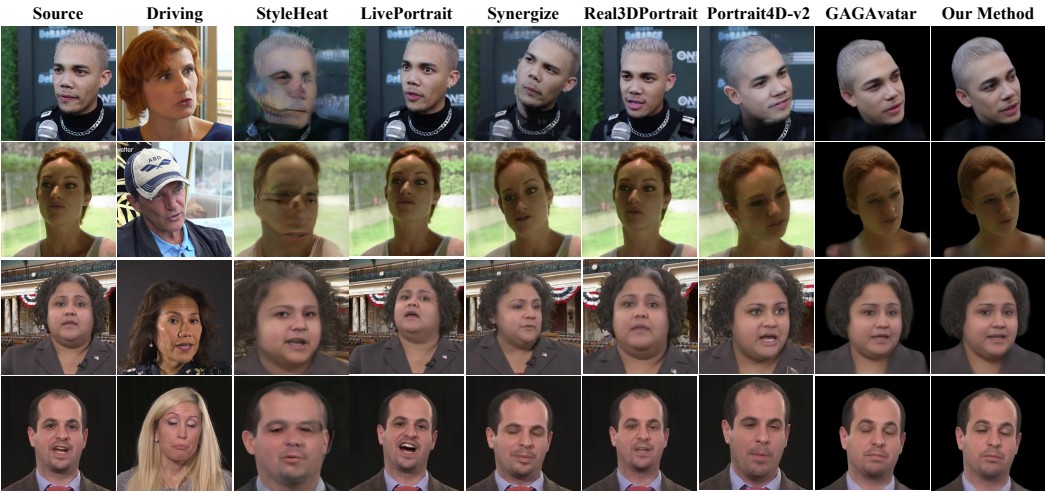

Figure 4: Cross-reenactment comparisons with state-of-the-art methods on VFHQ (rows 1–2) and HDTF (rows 3–4) datasets.

### 4.2.2  AUDIO-DRIVEN TALKING-HEAD GENERATION

Quantitative evaluation on image quality, video quality, and lip-synchronization between our Gen-FaceTalk and state-of-the-art audio-driven approaches are shown in Tab. 1. Our method achieves the best performance across all metrics, except for the Sync score on the VFHQ dataset. We also show qualitative comparisons in Fig. 3.

### 4.2.3  VIDEO-DRIVEN TALKING-HEAD GENERATION

Tab. 2 presents the quantitative results for video-driven generation between GenFaceTalk and the baselines. Our method consistently outperforms existing approaches on all metrics, with only minor exceptions in AED and APD under a few experimental settings. We believe this is primarily caused by the intrinsic jitter of the pose parameters estimated by the DECA detector (Feng et al., 2021). Furthermore, the FVD scores reported in our paper provide additional evidence for the temporal smoothness of the generated videos. We also show qualitative results in Fig. 4.

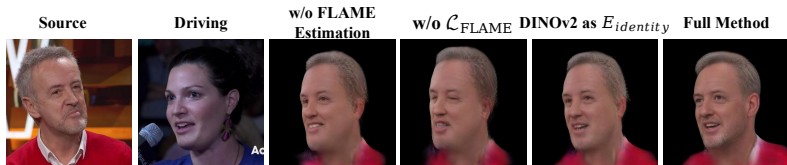

Figure 5: Visualization of ablation study results.

Table 3: Ablation results on the VFHQ dataset in self-reenactment setting.

| Method | PSNR ↑ | SSIM ↑ | FVD ↓ | AED↓ | APD↓ |
|---|---|---|---|---|---|
| w/o FLAME Estimation | 25.45 | 0.872 | 97.53 | 0.0152 | 2.73E-3 |
| w/o $\mathcal{L}_{FLAME}$ | 21.22 | 0.757 | 152.53 | 0.0377 | 1.27E-2 |
| DINOv2 as $E_{identity}$ | 23.14 | 0.781 | 83.07 | 0.0223 | 2.64E-3 |
| **Ours (full method)** | 25.96 | 0.895 | 48.96 | 0.0106 | 1.88E-3 |

## 4.3 ABLATION STUDY

### 4.3.1 EFFECT OF DIRECT FLAME ESTIMATION FROM DRIVING VIDEO

We investigate the importance of predicting FLAME parameters directly from the driving video rather than relying on offline 3DMM-based estimators. This design is critical for capturing temporally consistent motion and accurate spatial positioning. To evaluate its contribution, we replace our FLAME prediction pipeline (i.e. $E_{img}$, $E_{mlp}$, and the FLAME composition $x_D^{FLAME}$) in the motion learning module (Sec. 3.1) with an offline 3DMM estimator and remove the soft FLAME code loss $\mathcal{L}_{FLAME}$ during training. As shown in the second row of Tab. 3 ("w/o FLAME Estimation"), this leads to a slight drop in performance, confirming the effectiveness of our direct estimation strategy. Fig. 5 presents qualitative results from the ablation study.

### 4.3.2 IMPACT OF SOFT FLAME CODE LOSS $\mathcal{L}_{FLAME}$

We further assess the role of the soft FLAME code loss by training the model without $\mathcal{L}_{FLAME}$. As shown in the third row of Tab. 3 ("w/o $\mathcal{L}_{FLAME}$"), removing this loss results in a substantial performance drop across all metrics, indicating its utility in stabilizing FLAME parameter learning.

### 4.3.3 EFFECT OF HIERARCHICAL IDENTITY ENCODER

To validate the design of our hierarchical identity encoder in the identity learning module, we replace $E_{identity}$ with a fixed pre-trained DINOv2 (Oquab et al., 2023) and use its extracted features as the source appearance representation $x_S^{facial}$. The results, reported in the row "DINOv2 as $E_{identity}$" in Tab. 3, indicate a clear drop in image quality and temporal stability, highlighting the importance of jointly learning hierarchical features optimized for identity-preserving talking-head synthesis.

## 4.4 GENERALIZATION TO DIVERSE FACIAL STYLES

A key strength of GenFaceTalk lies in its ability to generalize beyond conventional human faces. To assess this, we conduct cross-identity reenactment experiments involving diverse facial styles, including animal faces and artistic portraits. As illustrated in Fig. 6, our model successfully preserves the structural and appearance characteristics of the source while accurately transferring motion attributes such as expression and head pose, demonstrating strong adaptability across domains.

## 4.5 MULTI-VIEW CONSISTENCY IN CROSS-IDENTITY REENACTMENT

To assess spatial consistency, we visualize multi-view reenactment results under novel camera poses in Fig. 7. Despite viewpoint changes, GenFaceTalk preserves identity fidelity and coherent facial motion, demonstrating robustness to diverse camera angles.

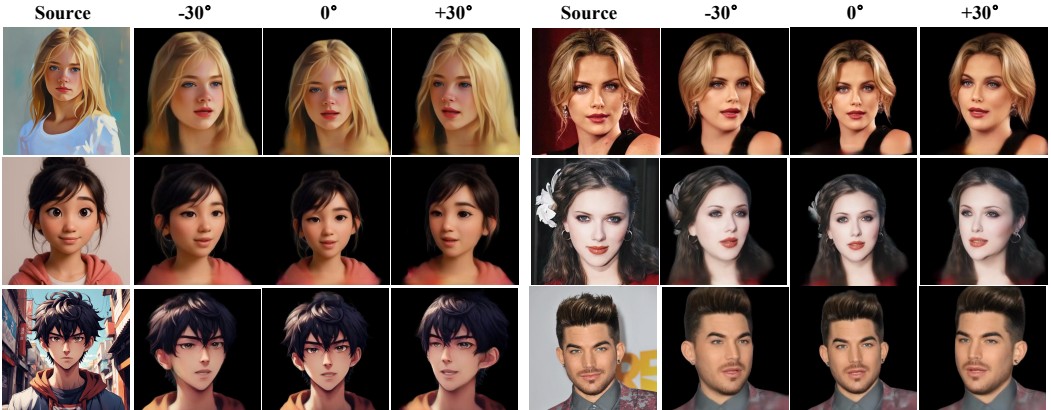

Figure 6: Reenactment results on stylized and non-human faces using our method. Our approach can handle faces across diverse styles while preserving appearance consistency, and ensuring that pose and expression faithfully follow the driving signal.

Figure 7: Multi-view reenactment results. Our method achieves high appearance and motion consistency across multiple views.

## 5  CONCLUSION

We present GenFaceTalk, a novel end-to-end one-shot talking-head generation framework based on 3D Gaussian Splatting. The core idea is to directly estimate FLAME parameters from the driving input via a learned encoder, rather than relying on offline 3DMM-based models. This design offers three key advantages: 1) it mitigates frame-wise inconsistencies and jitter commonly caused by inaccurate FLAME fitting in 3DMM pipelines. 2) It eliminates the need for offline preprocessing during inference, enabling a truly end-to-end and one-shot synthesis pipeline. 3) It generalizes beyond conventional human avatars, supporting a wide range of stylized and non-human facial domains. We further propose a joint learning strategy that fuses motion priors with hierarchical appearance cues, enabling temporally consistent, and identity-preserving avatar animation. Extensive experiments demonstrate that GenFaceTalk achieves superior performance in visual quality, temporal stability, and cross-domain generalization compared to state-of-the-art methods.

**Limitations.** Although our method removes the need for 3D face reconstruction at inference, it still relies on FLAME and the accuracy of offline supervision, which may limit performance under certain conditions. Extending the approach to leverage self-supervised or more expressive motion priors represents a promising direction for future work. Additionally, future research will explore incorporating multi-modal cues and improving robustness under challenging conditions, such as extreme poses and occlusions.

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
