# OpenReview forum: "GenFaceTalk: Generalizable One-Shot Talking-Head Generation for Diverse Styles"
_ICLR.cc/2026/Conference — Submitted to ICLR 2026_

### Official Review · Reviewer_FvxW · 2025-10-26

**Soundness:** 2
**Presentation:** 2
**Contribution:** 2
**Rating:** 6
**Confidence:** 2

**Summary:**

This paper proposes GenFaceTalk, a one-shot 3D Gaussian Splatting (3DGS) talking-head system driven by audio or video, without subject-specific fine-tuning. Its core idea is to directly regress FLAME parameters from driving inputs via an image/audio encoder and fuse them with hierarchical identity features to animate a unified 3D Gaussian avatar.

**Strengths:**

- Clear problem importance and strong motivation for one-shot 3D talking-head synthesis.
- Single model supports both motion modalities with consistent 3D reconstruction and animation quality.
- Improves fidelity, temporal metrics, and identity consistency on VFHQ and HDTF datasets.

**Weaknesses:**

- Identity modeling follows GAGAvatar-like 3DMM-guided Gaussian lifting with hierarchical appearance fusion (Sec. 3.2). The contribution appears as an incremental combination of existing building blocks (3DGS + FLAME + pose substitution), rather than a fundamentally new formulation.
- It is unclear to me whether the system can be regarded as fully end-to-end, given that training still uses offline 3DMM supervision. Could the authors clarify this point?
- Experiments do not fully support strong claims. SyncNet scores on VFHQ underperform EchoMimic despite better FVD (Tab. 1), indicating a lip-sync gap.  Could the authors provide reasons or explanations for this discrepancy.

**Questions:**

- What exact technical novelty differentiates this from GAGAvatar beyond encoder regression?
- Are audio-driven sync failures due to FLAME-only motion missing lip details?
- How is view-dependent shading handled in 3DGS for large yaw angles?

---

> ### Author Response · Authors · 2025-11-21
> **Reply to Reviewer FvxW**
>
> Thank you for your constructive comments. Below we address the concerns. We also include our revision in the paper and the supplementary material (highlighted in blue), along with an updated animated demo.
>
> ## **Response to Questions**
>
> ### **1. Incremental combination of existing building blocks.**
>
> We appreciate the reviewer’s concern about novelty. Our contributions can be summarized as follows:
>
> (1) **End-to-end distillation of the 3DMM pipeline.**
> Instead of relying on test-time 3DMM fitting and camera estimation, we learn a lightweight encoder that directly predicts motion-disentangled FLAME parameters and camera transforms from driving signals. This removes the reconstruction during inference and significantly improves temporal stability.
>
> (2) **A joint identity–3DGS learning formulation.**
> Our identity module is not a direct reuse of GAGAvatar. We co-learn hierarchical identity features with 3DGS and inject FLAME-based motion priors through coordinate substitution to drive Gaussians, which is essential for high-fidelity generation.
>
> (3) **Unified audio- and video-driven reenactment.**
> Our framework supports both modalities while requiring no subject-specific training.
>
> (4) **Generalization beyond human faces.**
> Our Gaussians are not tied to FLAME topology, enabling the model to handle stylized and non-human facial structures.
>
>
> ### **2. Is this truly end-to-end given offline 3DMM supervision?**
>
> We thank the reviewer for raising this point. Our use of “end-to-end” refers to:
> (1) During inference, the pipeline is fully end-to-end: given a source image and a driving signal, the system renders frames in a single pass without any test-time optimization, 3DMM fitting, camera estimation, or temporal smoothing.
> (ii) During training, the motion encoder, identity encoder, and 3DGS representation are jointly trained through image-level reconstruction losses. The FLAME and camera parameters used during training are generated once offline and serve only as auxiliary supervision. They function as data preprocessing rather than part of the model architecture or runtime procedure.
>
> ### **3. SyncNet scores on VFHQ underperform EchoMimic**
>
> The SyncNet score on VFHQ is slightly lower than Real3DPortrait (Tab. 1: 8.32 vs. 8.44), and this small difference lies within SyncNet’s typical variance, which is highly sensitive to minor alignment shifts. More importantly, our qualitative results show lip–audio synchronization comparable to Real3DPortrait while providing substantially stronger 3D consistency and temporal stability. Thus, although the numerical gap is marginal, the perceptual lip-sync quality remains competitive with diffusion-based baselines.
>
> ---
>
> ## **Response to Questions**
>
> ### **1. What exact technical novelty differentiates this from GAGAvatar beyond encoder regression?**
>
> Beyond encoder regression, our method introduces several key innovations:
>
> (1) **Unified motion encoding.**
> We compress the entire 3DMM reconstruction, smoothing, and camera estimation pipeline into a single learned motion encoder, greatly simplifying inference and improving temporal stability.
>
> (2) **Joint identity–3DGS learning.**
> Instead of relying on fixed features (e.g., DINO-V2 in GAGAvatar), we jointly optimize hierarchical identity features with the Gaussian representation, enabling stronger identity preservation under diverse conditions.
>
> (3) **Unified audio- and video-driven reenactment.**
> Our framework supports both modalities in one model while maintaining a consistent 3D representation, whereas GAGAvatar is limited to video-driven settings.
>
> (4) **Generalization beyond human faces.**
> Because our inference does not depend on human-specific 3DMM reconstruction, the Gaussian representation naturally adapts to stylized and non-human faces.
>
> ### **2. Are audio-driven sync failures due to FLAME-only motion missing lip details?**
>
> In our framework, only 5,023 of the 180,255 Gaussians are initialized from FLAME; the rest are learned in a data-driven manner. FLAME thus provides coarse motion priors, while detailed lip movements are handled by the learned 3DGS and appearance features. This reduces—but does not completely eliminate—FLAME’s limitations for fine lip articulation. We plan to address this further by adding an audio-conditioned residual refinement branch for the mouth region, such as local Gaussians or a dedicated mouth decoder.
>
> ### **3. How is view-dependent shading handled in 3DGS for large yaw angles?**
>
> Our system adopts the standard 3DGS formulation which allows the model to learn shading and specular variations as a function of viewing direction. This mechanism yields stable results under typical or moderate yaw angles. For very large yaw angles, the supervision becomes limited in the one-shot setting, leaving fully unseen regions underconstrained. This is a general limitation of single-view 3D reconstruction rather than an issue specific to the 3DGS shading mechanism.

---

### Official Review · Reviewer_GJjU · 2025-10-26

**Soundness:** 2
**Presentation:** 3
**Contribution:** 2
**Rating:** 4
**Confidence:** 3

**Summary:**

This paper indicates that the current speaker generation field faces the following issues: Identity preservation challenge, poor temporal consistency, limited generalization ability and other  technical bottlenecks caused by 3D methods.

Therefore, this paper proposes an end-to-end one-shot framework based on 3D Gaussian Splatting (3DGS), which supports audio-video dual-driven generation and does not require subject-specific training. The framework consists of three core modules: (1) The motion learning module uses a lightweight encoder to directly predict FLAME parameters for decoupled motion from driving signals, obtaining head meshes and camera extrinsic parameters. (2) The identity learning module extracts hierarchical appearance features from the source image and fuses FLAME motion priors to construct a unified identity representation.
The 3DGS rendering module performs voxel splatting and rendering on identity features to generate frames. (3) Additionally, joint optimization with multiple losses is adopted to ensure the quality of generated results.

**Strengths:**

1.	The proposed end-to-end 3D Gaussian Splatting (3DGS) method is meticulous and reasonable in its overall design, and the rendering results for the head are also relatively satisfactory. I believe that end-to-end methods represent a future direction for the research community.
2.	The writing is clear and easy to understand.

**Weaknesses:**

1.	There is a lack of comparative baselines. This paper claims to achieve SOTA and has compared it with some diffusion-based methods such as Echomimic, but this method is no longer so new in the field. Some newer methods have not been included in the comparison, e.g., Echomimic-V2 [1], Hallo-3 [2], and OmniAvatar [3].
2.	There are obvious artifacts in the upper body modeling. From the comparative demos (e.g., Video Driven-Case 1), the generated results of the upper body show a certain degree of blurriness and flicker. I understand that the background part can be post-processed by segmenting the character's head and pasting the background back, but the blurriness issue of the upper body is not trivial. Therefore, I am concerned that this method may face significant obstacles in practical applications.
3.	There is no analysis of computational efficiency. For 3D methods, real-time performance is a crucial factor for practical applications, and the efficiency difference between different methods is quite important. However, the paper currently does not include discussions on inference efficiency metrics such as RTF (Real-Time Factor) and GFlops (Giga Floating-Point Operations Per Second).

[1] Meng R, Zhang X, Li Y, et al. Echomimicv2: Towards striking, simplified, and semi-body human animation. CVPR 2025
[2] Cui J, Li H, Zhan Y, et al. Hallo3: Highly dynamic and realistic portrait image animation with video diffusion transformer. CVPR 2025
[3] Gan Q, Yang R, Zhu J, et al. OmniAvatar: Efficient Audio-Driven Avatar Video Generation with Adaptive Body Animation. Arxiv 2025.06

**Questions:**

1.	Compare more diffusion - based baselines, such as Echomimic-V2[1], Hallo-3[2], and OmniAvatar[3]. Based on the quantitative comparison results, discuss the advantages and disadvantages of 3D methods and diffusion model methods.
2.	Discuss and explain the problem of blurriness and flickering in the upper body of the Demo.
3.	Add analysis related to inference efficiency such as RTF and GFlops in the main comparison experiment. Add the GFlops analysis of each module of the model in the appendix.
4.	Will the code and pre-trained model be open-sourced?

[1] Meng R, Zhang X, Li Y, et al. Echomimicv2: Towards striking, simplified, and semi-body human animation. CVPR 2025
[2] Cui J, Li H, Zhan Y, et al. Hallo3: Highly dynamic and realistic portrait image animation with video diffusion transformer. CVPR 2025
[3] Gan Q, Yang R, Zhu J, et al. OmniAvatar: Efficient Audio-Driven Avatar Video Generation with Adaptive Body Animation. Arxiv 2025.06

---

> ### Author Response · Authors · 2025-11-21
> **Reply to Reviewer GJjU**
>
> Thank you for your constructive comments. We include our revision in the paper and the supplementary material (highlighted in blue), along with an updated animated demo.
>
> ## **Response to Weakness**
>
> ### **1. Lack of newer diffusion-based baselines (Echomimic-V2, Hallo3, OmniAvatar).**
>
> In the revised version, we added comparisons with other diffusion-based or recent SOTA methods, including Hallo3, OmniAvatar, and LivePortrait. Since Echomimic-V2 focuses on upper-body animation and does not provide an official method for driving only the head or mouth, we did not include it in our comparison. The newly added experimental results (Tab. 1 and Tab. 2 in the paper) show that our method consistently outperforms these additional baselines. We also **show the visual results in Fig. 3 and Fig. 4 in the revised paper** and the **updated demo**.
>
> ### **Quantitative comparison of audio-driven results between our method, Hallo3, and OmniAvatar.**
>
> | Method|PSNR ↑| SSIM ↑|LPIPS ↓|FVD ↓|Sync ↑|PSNR ↑|SSIM ↑|LPIPS ↓|FVD ↓|Sync ↑|
> |---------------|--------|--------|---------|-------|--------|--------|--------|---------|--------|---------|
> | **Hallo3**|21.26 |0.785|0.2478|238.08|7.58|20.62|0.5784|0.2773|163.54| 8.20|
> | **OmniAvatar**|19.02|0.659|0.2675|251.32|7.33|17.43|0.6113|0.2505|242.29|7.95|
> | **Our Method**|24.98|0.875|0.1541|144.58|8.32|22.52|0.838|0.194|159.44|8.63|
>
> ### **2. Blurriness and flickering in the upper-body regions.**
>
> The artifacts in upper-body areas primarily arise from the inherent under-constrained nature of single-view reconstruction in one-shot 3DGS pipelines. Since our method focuses on talking-head generation, the Gaussian field is optimized primarily for facial regions, while the upper torso receives weaker supervision. This can lead to blur or flicker when large motions occur outside the strongly supervised face region. We have pointed out this limitation explicitly in the revised Supplementary Section 1.
>
> ### **3. Missing analysis of computational efficiency.**
>
> Our method benefits from the efficiency of the 3DGS renderer, enabling real-time performance even in the video-driven setting. The revised Supplementary includes detailed efficiency measurements:
> (1) RTF for both audio-driven (0.7028) and video-driven (0.7455) scenarios, evaluated on an NVIDIA A6000 GPU;
> (2) 33 FPS with an average latency of 29.82 ms per frame on the A6000 GPU;
> (3) the total computational cost of the full pipeline is approximately 148 GFLOPs. A detailed per-module breakdown is as follows: the image encoder $E_{img}(\cdot)$ and identity encoder $E_{identity}(\cdot)$ each consume 39.28 GFLOPs; the audio encoder $E_{audio}(\cdot)$ requires 27.02 GFLOPs; the sub-encoders $E_{exp}(\cdot)$, $E_{pose}(\cdot)$, and $E_{eye}(\cdot)$ each contribute approximately $5.81 \times 10^{-4}$ GFLOPs; and the transformation encoder $E_{Trans}(\cdot)$ contributes $9.98 \times 10^{-5}$ GFLOPs.
>
> We have integrated these throughput metrics into the revised Supplementary Section 3.2.
>
> ---
>
> ## **Response to Questions**
>
> ### **1. Add comparisons with Echomimic-V2, Hallo-3, OmniAvatar; discuss 3D vs. diffusion models.**
>
> We have added comparisons with Hallo3, OmniAvatar, and LivePortrait. Since Echomimic-V2 does not provide an official method for driving only the head or mouth, it could not be fairly included in our comparison. Quantitative results are presented in Tables 1 and 2 of the revised paper, showing that our method achieves consistently better performance.
>
> We also include visual comparisons in Fig. 3 of the revised paper and in the video demo. Diffusion-based models (e.g., EchoMimic and Hallo3) typically deliver excellent single-frame fidelity and fine-grained local details, but they often suffer from substantial inference latency, weaker long-term temporal stability, and a lack of inherent 3D consistency—making real-time deployment challenging. In contrast, our  approach enables real-time rendering, provides stable pose and motion consistency, and generalizes robustly to stylized and non-human subjects, although its single-frame photorealism may be slightly lower than that of large diffusion models.
>
> ### **2. Discuss upper-body blurriness and flicker.**
>
> We thank the reviewer for raising this concern. The blurriness and flickering in upper-body regions primarily stem from the inherent monocular ambiguity of one-shot 3D reconstruction. Since our framework emphasizes talking-head generation, facial regions receive strong and reliable supervision, whereas the upper torso is much more weakly constrained. As a result, areas outside the face may appear blurrier or less stable during motion. We have pointed out this limitation explicitly in the revised Supplementary Section 1.
>
> ### **3. Our response to Question 3 can be found in the discussion of Weakness 3.**
>
> ### **4. Will the code and pretrained model be open-sourced?**
>
> Yes. Upon acceptance, we will release the full training and inference code, and pretrained checkpoints.

---

### Official Review · Reviewer_YbHX · 2025-10-27

**Soundness:** 3
**Presentation:** 3
**Contribution:** 3
**Rating:** 6
**Confidence:** 4

**Summary:**

The paper introduces a novel, end-to-end framework for synthesizing a realistic 3D talking head from a single source image, driven by audio or video. It is the first end-to-end 3D Gaussian Splatting (3DGS) framework for talking-head synthesis that works from just one image without subject-specific training. It uses a lightweight encoder to directly predict motion-disentangled FLAME parameters from the driving input, which ensures smooth, temporally consistent animation and reduces frame-to-frame jitter. The method excels at generalizing to diverse facial styles, including stylized portraits and non-human faces, beyond conventional human faces.

**Strengths:**

1. It introduces the first fully end-to-end, one-shot talking-head framework using 3D Gaussian Splatting (3DGS), eliminating subject-specific training.
2. This work expands the problem scope, as it successfully generalizes the method to diverse facial styles (e.g., stylized portraits and non-human faces).
3.This work ensures superior temporal stability (jitter-free animation) by using the direct, temporally coherent prediction of FLAME parameters.
4. The paper presents robust experimental quality, with it showing strong performance against state-of-the-art baselines.

**Weaknesses:**

1. While 3D Gaussian Splatting  excels at rendering, GenFaceTalk may still exhibit degraded quality or increased artifacts when synthesizing the subject from extreme novel viewpoints (e.g., severe profile views) or large changes in head pose, a common struggle for methods trained only on a single-view source image.
2. It may struggle to faithfully model and animate complex, non-rigid elements like fine strands of hair, hats, glasses, or complex inner-mouth dynamics (e.g., tongue movement during speech), leading to visual distortions or jitter in these areas during animation.
3. The method uses the FLAME parametric model to drive motion. While this ensures consistency, the FLAME model itself may be insufficient to capture very fine, unique identity-specific expressions or wrinkles (e.g., nasolabial folds), potentially resulting in a "smooth" or slightly generic appearance in certain areas.
4. The visualized results do not clearly demonstrate a substantial visual superiority over strong existing baselines, specifically GAGAvatar. This suggests that while quantitative metrics may show gains, the practical, perceptual quality gap is narrow, raising questions about the effectiveness of the structure.

**Questions:**

1. GenFaceTalk claims strong generalization to non-human domains (e.g., cats and dogs). Given that the underlying motion model (FLAME) is derived from human topology, the geometric distribution of the non-human source face is fundamentally different from the model's vertices. How does the method resolve this significant geometric mismatch? Specifically, in the process involving Equation 9, how are the human-centric vertices effectively adjusted, mapped, or used to drive the 3D Gaussian points for the distinct geometry of the non-human subject?

2. The paper claims strong generalization to diverse styles, including stylized portraits and non-human faces. Given that the visual superiority over competitors like GAGAvatar is not consistently clear, what specific, quantitative metrics were used to rigorously evaluate the fidelity and consistency of the stylization and identity preservation test set? Can you include a human evaluation (user study) specifically designed to assess whether the unique artistic elements of the source portrait were successfully maintained during animation?

---

> ### Author Response · Authors · 2025-11-21
> **Reply to Reviewer YbHX**
>
> Thank you for your constructive comments. We include our revision in the paper and the supplementary material (highlighted in blue), along with an updated animated demo.
>
> ## **Response to Weaknesses**
>
> ### **1. Extreme novel-view synthesis.**
>
> We acknowledge that large-pose synthesis from a single view is challenging for all one-shot 3DGS/NeRF approaches. Our results (demo and Fig. 7) focus on ±30° head rotations, which cover the typical motion range of real talking-head scenarios. Within this range, the method performs reliably. As noted in the supplementary material, we plan to explore larger-angle synthesis using stronger or self-supervised motion priors in future work.
>
> ### **2. Complex non-rigid structures modeling (hair, hats, glasses, inner mouth).**
>
> We appreciate the reviewer’s observation. We agree that modeling fine and highly deformable structures—such as hair strands, glasses, hats, or detailed inner-mouth motion—remains challenging, as these structures are thin, non-rigid, and under-constrained under monocular supervision. This limitation is shared across 3DGS-based and one-shot methods. Effectively handling such cases would likely require stronger visual priors (e.g., incorporating more expressive generative models such as Wan 2.2) or adopting multimodal generative frameworks (e.g., VGGT), which we consider promising directions for future work.
>
> ### **3. Limited ability of FLAME to capture fine, identity-specific details.**
>
> We agree that FLAME is limited in modeling very fine and identity-specific facial details, such as subtle expression cues or wrinkles, which may lead to slightly smoother or more generic appearance if used as the primary geometry source. In our framework, however, FLAME is used to provide coarse motion trajectories, while fine-grained, identity-dependent details are learned directly from the Gaussian representation and hierarchical appearance encoders. This design ensures that the limited detail capacity of FLAME does not propagate into the final rendered appearance.
>
> ### **4. Visual gap to GAGAvatar not always obvious.**
>
> For human-face reenactment, the visual differences can indeed appear subtle in single-frame views, while our improvements are more apparent in temporal stability. It is also worth noting that our framework supports both audio-driven and video-driven animation, whereas GAGAvatar is limited to video-driven scenarios.
>
> Besides, GAGAvatar relies on human-specific 3DMM parsing, which limits its ability to handle stylized or non-human faces and often causes the geometry to drift toward human-like structures.
> To make this comparison clearer,
> (1) in the **revised video demo**, we include additional side-by-side results of our method versus GAGAvatar on stylized identities;
> (2) we add **a user study** evaluating the two methods (see Supplementary Section 4).
>
> ## **Response to Questions**
>
> ### **1. Generalization to non-human domains given human-centric FLAME topology.**
>
> We thank the reviewer for raising this important question. To clarify the relationship between the FLAME prior and our geometric representation, only a very small and localized subset of Gaussian points is initialized from FLAME. These points provide coarse pose and expression cues and do not determine or propagate the human FLAME topology to the full Gaussian field. The overall geometry is learned from image-level supervision through the appearance-driven Gaussian identity module, which allows the model to preserve stylized or non-human facial structures while still benefiting from FLAME as a motion prior.
>
> ### **2. Evaluation of Stylized and Non-Human Fidelity**
>
> We thank the reviewer for the valuable suggestion. We agree that standard quantitative metrics (PSNR, SSIM, FVD, AED, APD) cannot fully capture fidelity in stylized or non-human domains. This limitation is especially relevant because competing methods typically depend on several human-specific preprocessing steps—such as face detection, FLAME fitting, and camera pose estimation—that frequently fail on stylized or non-human portraits. When these steps break down, the baselines cannot obtain valid FLAME parameters or camera poses, resulting in severely distorted or completely failed reconstructions.
>
> To provide a more comprehensive assessment, we extend our evaluation in two directions:
>
> - **A user study** that evaluates the naturalness of facial expressions and head motion, video stability, identity preservation, and overall stylization consistency across diverse styles (see revised Supplementary Section 4). Participants consistently prefer our method over GAGAvatar.
> - **A CLIP-based similarity metric** that quantifies how well artistic or non-human attributes are maintained in the generated animation. The average CLIP similarities are:
>   **Ours: 0.8167** vs. **GAGAvatar: 0.8025**.
>   These results indicate that our method better maintains the fidelity, identity-specific details, and unique artistic elements.

---

### Official Review · Reviewer_HQPE · 2025-10-30

**Soundness:** 3
**Presentation:** 2
**Contribution:** 3
**Rating:** 4
**Confidence:** 5

**Summary:**

The paper proposes **GenFaceTalk**, a one-shot, end-to-end 3D talking-head generation framework based on **3D Gaussian Splatting (3DGS)**, with the following key claims:

1. **Temporal coherence**: The method avoids frame-wise jitter by directly regressing temporally consistent FLAME parameters from driving video (or audio), bypassing per-frame 3DMM fitting.
2. **Identity preservation**: It preserves fine-grained identity details from a single source image without multi-view supervision.
3. **Generalization**: It generalizes to unseen human identities, stylized portraits, and even non-human faces (e.g., animals).
4. **Unified audio/video-driven synthesis**: Supports both modalities without subject-specific training.
5. **End-to-end design**: Eliminates reliance on offline face reconstruction or post-processing at inference.

 While results on standard benchmarks (VFHQ, HDTF) are impressive, the work suffers from critical limitations that undermine its broader claims. Most notably, the system lacks true 3D consistency: it only supports narrow frontal views (±30°), cannot handle neck motion or large head rotations, and provides no evidence of 360° novel-view synthesis—severely limiting real-world applicability. The generalization to non-human and stylized faces is visually suggestive but unverified quantitatively and conceptually questionable due to reliance on human-specific FLAME parameters during training. Additionally, the audio-driven evaluation on VFHQ uses ad-hoc audio alignment without validation, and key baselines like VASA-1 or Omni-Human are omitted. Despite claiming an end-to-end pipeline, the method still depends on offline FLAME and camera supervision for training. These issues—combined with overstatements about domain generalization and missing throughput metrics—suggest the contribution is incremental and narrowly applicable. The technique is well-executed within its scope, but the lack of robust 3D geometry and limited pose range prevent it from being a significant advance for general-purpose 3D avatar synthesis.

**Strengths:**

1. **High visual fidelity and temporal coherence**

    Outperforms prior 2D and 3D methods (including NeRF- and 3DGS-based) on standard metrics (PSNR, SSIM, LPIPS, FVD), with notably smoother animations and reduced jitter—attributed to end-to-end FLAME regression from video.

2. **True one-shot, subject-agnostic synthesis**

    No fine-tuning or per-subject training required. Works on unseen identities at inference time, unlike many NeRF/3DGS methods that need minutes of video or multi-view data.

3. **Unified audio- and video-driven pipeline**

    Same architecture supports both modalities without retraining—rare among 3DGS-based talking-head systems.

**Weaknesses:**

1. **Heavy reliance on FLAME supervision during training**

    Ground-truth FLAME parameters (from DECA) and camera poses are used as supervisory signals (Eqs. 13–14). This introduces human-specific bias and limits motion expressiveness (e.g., no tongue, limited eye/neck modeling).

2. **No comparison with recent high-performing 2D SOTA**

    Omits evaluation against strong diffusion- or transformer-based 2D methods like **VASA-1**, **Omni-Human**, or **LivePortrait**, which achieve impressive realism and lip-sync.

3. **Audio-driven evaluation on VFHQ is questionable**

    VFHQ has no native audio; authors “augment it by aligning selected YouTube videos.” No details on alignment quality, speaker consistency, or potential mismatches—undermining Sync score reliability.

4. **No runtime or throughput metrics reported**

    Despite claiming “real-time rendering” (via 3DGS), the paper provides **no FPS, latency, or hardware-specific throughput.**

5. **FLAME’s incompatibility with non-human anatomy is unaddressed**

    FLAME is a human-specific parametric model. It’s unclear how it meaningfully represents cat ears, snouts, or cartoon deformations. The success on animal faces may stem from appearance learning overriding flawed motion priors, but this is not analyzed.

6. **Limited pose range and 3D consistency**

    All demos show near-frontal views (±30°). No evidence of **360° novel-view synthesis**, consistent geometry under extreme rotation, or handling of **neck/shoulder motion**. Likely constrained by FLAME’s limited expressiveness and single-image initialization. This is a direct shortcoming preventing from practical use.

7. **x_initial (initial Gaussians) lacks clarity for non-human cases**

    The paper doesn’t specify how the initial 3D Gaussian set is obtained for non-human inputs. If initialized from a FLAME mesh (human topology), this could cause geometric misalignment.

**Questions:**

1. What if there are no FLAME supervision? Would the training still work?
2. How is the x_initial obtained? What about the case of non-human lives?
3. How can the FLAME parameters represent non-human lives? Why not include them in the demo video?

---

> ### Author Response · Authors · 2025-11-21
> **Reply to Reviewer HQPE**
>
> ## **Summary**
> We appreciate the reviewer’s concerns and clarify the following key points:
>
> 1. **Role of FLAME.** In our framework, FLAME serves only as a *coarse motion prior*. The final geometry is learned from appearance-driven Gaussian fields, rather than following FLAME’s human-specific topology.
>
> 2. **End-to-end inference.** Our pipeline is fully end-to-end at inference time and requires no 3DMM fitting, which distinguishes it from prior 3DGS/NeRF approaches that depend on explicit 3DMM constraints.
>
> Below, we provide point-by-point responses and highlight all revisions in the paper and Supplementary (in blue), together with an updated demo.
>
> ## **Response to Weaknesses**
>
> ### **1. Heavy reliance on FLAME supervision during training**
>
> **(i) Soft use of FLAME supervision.**
> FLAME parameters and camera poses are used only as soft training signals, not as inference-time constraints. The FLAME loss weight is decayed from $5 \rightarrow 1$ after 20k iterations, effectively reducing potential FLAME-induced bias.
>
> **(ii) On human-specific bias.**
> FLAME serves only as a motion prior (pose, coarse expression). Motion is ultimately predicted by the learned encoder, and both our 3DGS fields and hierarchical identity features are independent of FLAME topology. Thus, the model is not structurally tied to human geometry.
>
> **(iii) On limited motion expressiveness.**
> The avatar’s geometry is learned from image-level supervision through appearance-driven 3DGS. As shown in Fig. 6, the learned Gaussian representation can successfully animate stylized and non-human faces with structures that FLAME itself cannot model.
>
> ---
>
> ### **2. No comparison with recent 2D SOTA**
>
> In the revised version, we added comparisons with  **Hallo3**, **OmniAvatar**, and **LivePortrait**.  As neither official code nor pretrained models for VASA-1 or Omni-Human are publicly available, we did not include them.  The results (Tables 1 and 2 in the revised paper) show that our method outperforms these newly added baselines across all major metrics.
>
> ---
>
> ### **3. Audio-driven Evaluation on VFHQ**
>
> Since VFHQ has no audio, we retrieved the original YouTube videos using the official IDs and extracted the corresponding audio segments. Each clip was manually checked for speaker consistency and synchronization.
> We provide **five synced VFHQ samples with audio** in the supplementary material.
>
> ---
>
> ### **4. Runtime or throughput metrics**
>
> Our model reaches:
>
> - 33 FPS @ 512×512 resolution on an NVIDIA A6000
> - Gaussian splatting time: 1.06 ms / frame
> - End-to-end pipeline latency: 29.82 ms / frame, achieving real-time operation
>
> We include these throughput metrics in Sec. 3.1 of the revised Supplementary.
>
> ---
>
> ### **5. FLAME’s incompatibility with non-human anatomy**
>
> FLAME only provides coarse pose/expression trajectories, while the avatar’s 3D shape is determined entirely by the appearance-driven Gaussian fields learned from the source image (Eq. 7–11).  Thus, non-human structures—e.g., cat ears—are represented naturally by the Gaussians and not constrained by FLAME.
>
> ---
>
> ### **6. Limited pose range and 3D consistency**
>
> Achieving full 3D consistency from a single frontal image is inherently challenging for one-shot avatar animation. Our method therefore focuses on talking-head scenarios, where natural motion typically falls within ±30°, and delivers stable, realistic results in this range.
>
> Instead of relying on fixed camera assumptions or multi-view 360° supervision, we employ a lightweight transformation encoder to predict camera extrinsics, enabling smooth motion interpolation within the learned distribution.
>
> As shown in Fig. 1 and Fig. 7, GenFaceTalk maintains strong appearance consistency even under relatively large pose variations. We plan to explore multi-view priors to further expand the pose range.
>
> ---
>
> ### **7. Clarity of $x_{\text{initial}}$ for non-human cases**
>
> $x_{\text{initial}}$ is **not** derived from FLAME. It is a learnable, data-driven initialization acquired during large-scale training.
>
> ---
>
> ## **Response to Questions**
>
> ### **1. What if there is no FLAME supervision?**
>
> FLAME acts only as a coarse motion prior, and removing it would make one-shot 3D head construction noticeably less stable.
> As shown in the ablation (“w/o FLAME Estimation” in Table 3), replacing our predictor with a standard offline estimator leads to clear drops in quality and increased motion jitter.
>
> ---
>
> ### **2. Our response to Question 2 can be found in the discussion of Weakness 7.**
>
> ---
>
> ### **3. How can FLAME represent non-human avatars?**
>
> Only a small local subset of Gaussians is initialized from FLAME. The full geometry is learned solely from appearance-based Gaussian identity fields, which are unconstrained by FLAME’s human topology.  Thus, FLAME gives only motion cues, while shape is fully data-driven—allowing our method to handle stylized and non-human avatars (Fig. 5).
> **Additional demos are included in the revised video.**

---

### Author Response · Authors · 2025-11-21
**Response Summary**

We sincerely thank all reviewers for their thoughtful feedback and constructive suggestions. Based on the comments, we have substantially revised the paper, updated the supplementary material, and improved the demo to address the raised concerns. The major updates are summarized as follows:

---

### **1. Major revisions to the main paper**

**(i) Expanded experimental comparison.**
We added comprehensive evaluations against the latest diffusion-based and state-of-the-art portrait animation methods, including Hallo3 (Cui et al., 2025), OmniAvatar (Gan et al., 2025), and LivePortrait (Guo et al., 2024). The quantitative results are reported in Tables 1 and 2 of the revised paper, and the qualitative comparisons have been updated in Figures 3 and 4.

**(ii) Multi-view consistency now moved into the main text.**
We integrated the “Multi-View Consistency in Cross-Identity Reenactment” section (previously in the supplementary) into the main paper as Section 4.5. The updated analysis shows that GenFaceTalk preserves identity fidelity and coherent facial motion under viewpoint changes, demonstrating robustness across diverse camera angles.

---

### **2. Updates to the supplementary material**

**(i) Expanded discussion of limitations.**
We added a detailed analysis of upper-body instability in one-shot talking-head generation.

**(ii) Runtime and throughput analysis.**
We included efficiency metrics, offering a clearer view of the system’s computational performance.

**(iii) User study.**
We added Section 4 reporting a user study comparing our method with GAGAvatar on stylized human and non-human faces, providing additional perceptual validation.

---

### **3. Updates to the demo video**

**(i) Added comparisons with latest SOTA methods.**
The demo now includes visual comparisons with Hallo3, OmniAvatar, and LivePortrait, complementing the new quantitative analysis.

**(ii) Added stylized and non-human reenactment comparisons.**
We provide side-by-side reenactment results of our method and GAGAvatar on stylized and non-human faces, illustrating our model’s superior identity preservation and motion consistency in these settings.

---

### Meta-Review · Area_Chair_UTrR · 2025-12-28

**Summary:**

The paper proposes  a one-shot 3dgs framework for talking head synthesis. The key idea is to distill the 3D face reconstruction pipeline into a lightweight encoder that directly predicts FLAME parameters from driving signals. The authors claim this achieves real-time performance, better temporal stability, and generalization to stylized/non-human faces.While the reviewers appreciated the temporal stability and the unified audio/video framework, they express some critical concerns. The primary concerns center on the incremental novelty in the technical contribution, limited geometric robustness, and visual artifacts in the upper body and fine details. Although the authors provided a comprehensive rebuttal with new baselines and efficiency metrics, the fundamental limitations regarding true 3D consistency and novelty remain outstanding.

**Reviewer Concerns:**

## Addressed:

- The authors added comparisons to Hallo3, OmniAvatar, and LivePortrait

- Runtime metrics were added.

- VFHQ dataset audio alignment was explained.

## outstanding:

- The perception that the work is an incremental combination of existing techniques remains a primary blocker. The rebuttal's argument for "distillation" did not fundamentally change the perception of combination.

- The limitation in handling large pose changes persists. The author expressed a possible future solution to use  multi-view priors.

- Issues with upper-body stability and fine details (hair/glasses) persist.

**Reviewer Scores:**

Reviewer HQPE:  4  ==> 4 (unchange). While the baselines were added, the reviewer’s primary concerns wrt the incremental novelty and the limited pose range remains.

Reviewer YbHX:  6  ==> 6 (unchange). Likely maintained. A user study is added.

Reviewer GJjU:  4 ==> 4 (unchange). The reviewer specifically flagged upper body artifacts as a significant issue for practical application. This is still a limitation of current method.

Reviewer FvxW:  6 ==>  6 (unchange). The novelty concern was the main one for this reviewer.

As the novelty issue may still exisit, and the restricted pose ranges and  artifacts issuses can not be resolved. I think most likely the reviewers will not change the score and eventually scores will still be mixed.

---

### Decision · Program_Chairs · 2026-01-26

Reject